# Evaluating the mutagenic potential of aerosol organic compounds using informatics based screening

Stefano Decesari[1*], Simona Kovarich[2]; Manuela Pavan[2]; Arianna Bassan[2]; Andrea Ciacci[2], David Topping[3,4]

[1]Institute of Atmospheric Sciences and Climate, National Research Council of Italy (ISAC-CNR), 40121, Bologna, Italy
[2]S-IN Soluzioni Informatiche Srl, 36100, Vicenza, Italy
[3]School of Earth, Atmospheric and Environmental Sciences, The University of Manchester, M13 9PL
[4]National Centre for Atmospheric Science, The University of Manchester, M13 9PL

*Correspondence to*: S. Decesari (s.decesari@isac.cnr.it)

**Abstract.** Whilst general policy objectives to reduce airborne particulate matter (PM) health effects are to reduce exposure to PM as a whole, emerging evidence suggests more detailed metrics associating impacts with different aerosol components might be needed. Since it is impossible to conduct toxicological screening on all possible molecular species expected to occur in aerosol, in this study we perform a proof of concept evaluation on the information retrieved from *in silico* toxicological predictions, in which a subset (N = 104) of secondary organic aerosol (SOA) compounds were screened for their mutagenicity potential. An extensive database search showed that experimental data are available for 13% of the compounds while reliable predictions were obtained for 82%. A multivariate statistical analysis of the compounds based on their physico-chemical, structural and mechanistic properties showed that 80% of the compounds predicted as mutagenic were grouped into six clusters, three of which (5-membered lactones from monoterpene oxidation, oxygenated multifunctional compounds from substituted benzene oxidation, and hydroperoxides from several precursors) represent new candidate groups of compounds for future toxicological screenings. These results demonstrate that coupling model-generated compositions to *in silico* toxicological screening might enable more comprehensive exploration of the mutagenic potential of specific SOA components.

## 1 Introduction

Ambient air pollution was ranked as the 7[th] highest risk factor for human health (Lim et al. 2012), being responsible for almost three billions deaths per year globally. Evidence for air pollution impacts on life expectancy and for cardiovascular and respiratory illnesses has grown considerably in the last two decades (Beelen et al., 2014), and the ongoing global demographic and societal changes (ageing, urbanization) are projected to exacerbate atmospheric pollution health effects. Evidence from both short and long-term epidemiological effects of particulate matter with particle diameter below 10 µm or 2.5 µm (PM10 or PM2.5, respectively) is robust, with a range of possible policies aiming to mitigate PM health effects reflected by the possible sources from which PM arises.

Since at least the early 2000's, metrics for PM chemical composition and sources have been incorporated along with PM10, PM2.5 or PM0.1 (ultrafine) in epidemiological studies. Recently, black carbon (BC), which is a proxy for primary combustion particles, was associated with an increased risk of mortality two times greater than for total PM (Janssen et al., 2011). Recent findings highlight that more detailed air quality metrics may be valuable in evaluating the health risks by specifically

distinguishing between e.g. black carbon, secondary organic and inorganic aerosols (Cassee et al., 2013). A study in London, UK, suggested that certain particle components might be more important to specific diseases (Atkinson et al., 2010), some toxicological studies suggesting pulmonary and vascular inflammation as the relevant biological response mechanism (WHO 2013; HEI 2010). Despite this, the overarching developments of policy objectives remain to substantially reduce population exposure to PM2.5 as a whole. Stanek et al. (2011) provided a compilation and meta-analysis of 29 epidemiological and in

vivo toxicological studies that explicitly investigated the statistical relationship between PM chemical composition and adverse health effects. They noted that, from a mechanistic perspective, it is highly plausible that the chemical composition of particulate matter (PM) would better predict health effects than other characteristics, such as PM mass or size (Stanek et al., 2011). However, their conclusion is that no consistent relationships have emerged so far: there is little evidence of a systematic association of specific groups of compounds with adverse effects across all studies. Possible reasons for the inconsistencies

between epidemiological results are the paucity of studies, non-additive (synergic) effects of pollutants and study design issues: differences in the exposed populations, differences in pollution levels between environments, and differences in the air quality networks. In particular, the sets of PM chemical compounds measured by different monitoring networks overlap for only a few species: metals, inorganic ions, BC, possibly organic carbon (OC), with little or null information of OC molecular composition. West et al. (2016) note that epidemiologic studies should consider other air pollutants suspected to be important

for health, such as polycyclic aromatic hydrocarbons (PAHs), metals, reactive oxygen species (ROS), and other chemical components of PM. These include classes of compounds defined by their reactivity (ROS) with unclear overlaps with the other compounds in the list (Zhang et al., 2008). More importantly, guidelines should be provided for the identification of possible targets (the compounds suspected to be important for health). A group of substances commonly monitored (PAHs and metals) has been extensively characterized by the toxicological research on combustion and industrial aerosols. On the other hand, the

recent advances in the chemical characterization of ambient organic particles, including secondary organic aerosols (SOA) (Hallquist et al., 2009), have been exploited in PM toxicology only to a limited extent (Verma et al., 2015; Lakey et al., 2016). Given emerging evidence points towards the need for more information on associations between health impact potential and separate PM fractions, it is worthwhile evaluating the role of emerging technologies in extracting this information.

At present, only information on bulk SOA in vivo and in vitro biological effects is available (Delfino et al., 2008; Chen et al.,

2011; Rohr et al., 2013; Saffari et al., 2015; Künzi et al., 2015). The lack of actual molecular targets for the SOA fraction responsible for its toxicity might pose limitations to the design of epidemiological studies incorporating specific SOA tracers. Moreover, while molecular identification of SOA compounds is normally achieved through the analysis of exact molecular masses and the interpretation of mass fragmentation spectra, the synthesis of authentic standards has been conducted to a very limited extent and for a few compounds. The lack of standards is critical for the determination of toxicity endpoints via classical

toxicological assays. Estimation of such endpoints can be pursued, however, on the basis of the sole chemical composition by means of *in silico* approaches, such as quantitative structure-activity relationships (QSARs). *In silico* toxicology is widely used in drug chemistry, especially as a screening before in vitro and in vivo assays. In particular, QSARs are theoretical models relying on a mathematical relationship (often a statistical correlation) between one or more quantitative parameters derived from chemical structure and a quantitative measure of a property or activity (e.g. a toxicological endpoint). The reliability of a QSAR prediction highly depends on the scientific validity of the employed QSAR model, which should fulfill the internationally recognized principles for QSAR validation as defined by OECD (OECD 2004, 2006, 2007), and the applicability domain of the QSAR model, i.e. whether the compound of interest is sufficiently similar to the compounds used to train the model (Netzeva et al., 2005; Sushko et al., 2010; Sahigara et al. 2012). So far, as noted previously, the use of *in silico* methods for the toxicological assessment of aerosol organic compounds has been limited and restricted to specific chemical classes like PAHs (Papa et al., 2008; Schwarz et al., 2014). In this study, we perform a proof of concept evaluation on the information retrieved from *in silico* methods specifically applied to a subset of measured compounds attributed to multiple sources of SOA. These compounds have been largely unexplored with regards to their toxicological properties, and the general evaluation of available *in silico* methods has hitherto not been carried out. The potential of such methods is considerable as, in principle, toxicity endpoints of aerosols can be generated directly from molecular compositions that can either be measured directly or predicted using state-of-the-art explicit chemistry models.

This evaluation is timely. In the past decade, the rapid development of aerosol mass spectrometric techniques of increasing resolution has boosted the molecular identification of SOA compounds (Laskin et al., 2012; Nozière et al., 2015). At the same time, mechanistic models predict the chemical composition of SOA with an increasing degree of sophistication by accounting for the movement of compounds between the gaseous and condensed phases at a molecular level. Despite uncertainties in fundamental parameters leading to variable mass and composition predictions (McVay et al., 2016), the amount of information available from both measurement and modelling technologies might be combined to start associating health impact potential with distinct groups of compounds. This is the driver for the study presented here.

## 2 Methods

A list of ~100 SOA compounds was identified for subsequent toxicity predictions using *in silico* (i.e. QSAR) methods. First, in vitro genotoxicity data as bacterial mutagenicity (Ames test) available for these compounds were searched for in several international databases. Following this, QSAR predictions for bacterial mutagenicity were performed for all the species, and the results were compared to the toxicity endpoints determined by in vitro methods (for the subset of compounds for which in vitro data could be found). Finally, the results of the toxicity predictions were interpreted using a cluster analysis of the compounds based on a suite of molecular descriptors: structural, mechanistic (in relation to express toxicity on humans), etc., for understanding inter and intra -class variability.

**2.1 Organic compounds**

A set of 104 compounds speciated according to biogenic and anthropogenic SOA was compiled for this proof of concept study (Table S1). They are organized in chemical classes defined as products originating from specific volatile organic compounds (VOCs). In order to keep the total number of compounds to ~100, a number of isomeric species have been filtered out. The list as a whole is an attempt to sample the greatly diverse chemical space of SOA species that can be derived from the vast literature on this subject. To create this list, exemplar molecular species were chosen to represent all relevant classes of SOA treated in recent review studies (Hallsquit et al., 2009; Nozière et al., 2015), which include the products of oxidation of isoprene, monoterpenes (α-pinene, β-pinene, Δ3-carene, *d*-limonene), sesquiterpenes, as well as of anthropogenic VOCs: substituted benzenes, alkenes and amines. SOA compounds from intermediate-volatility organic precursors ("IVOCs": low-molecular weight PAHs, and $C_{12}$-$C_{16}$ *n*-alkanes) and examples of the products of heterogeneous (particle-phase) reactions of small carbonyls (glyoxal, methyl-glyoxal, pyruvic acid) are also included. Finally, highly-reactive compounds forming from radical chemistry, such as secondary ozonides, adducts of carbonyls on Criegee Intermediates (CI), and polyperoxides ("highly oxygenated organic molecules", HOMs) are accounted for. A range of secondary organic compounds missing an unequivocal source assignment, such as oxalic or succinic acid, were excluded.

The list in Table S1 includes compounds of varying degrees of "identification factor" ("I factor", (Nozière et al., 2015)): for only a fraction of them a chromatographic separation, confirmation with standards and quantification in real samples could be achieved. These include, for instance, most common α-pinene SOA tracers, including pinonic, hydroxyl-pinonic and pinic acids (mtr_05, mtr_06 and mtr_07 in Table S1), terpenyllic, terebic and diaterpenyllic acids (mtr_11, mtr_12 and mtr_13), 3-methyl-1,2,3-butanetricarboxylic (MBTCA) (mtr_17), together with their dimers (mtr_15, mtr_16), which can add up 10 to 15% of SOA mass (Kristensen et al.; 2014). Other well-known SOA compounds included in the list account for the isoprene chemistry: methyl-tetrols (iso_01 in Table S1), C5-alkene triols (iso_04) and their sulfate esters (iso_02) are common constituents of organic aerosols in rural areas, where they can account for 12 – 14% of ambient particulate organic matter (Lin et al., 2013). Concentration data for anthropogenic compounds, such as quinones, oxo- and nitro-PAHs are also available in the literature (Chung et al., 2006; Huang et al., 2014). For most of the other organic compounds in the list, concentration data are simply not available. These include molecular species of uncertain identification, and whose molecular structures were inferred solely from the interpretation of mass spectra. It should be emphasized that this compound list was similarly formulated in order to maximize the diversity between molecular structures. As a consequence, the chemical classes that correspond to the most extensively studied laboratory SOA systems have more representative compounds than the SOA systems for which only a limited sets of conditions (e.g., photochemical) have been explored. Overall, the list of compounds was not designed to reproduce the entire molecular composition of SOA in any particular environment. In this sense, and to re-iterate the rationale behind this study, we use this list of compounds to demonstrate the information that can be extracted from an *in silico* assessment of the mutagenic activity of the main classes of anthropogenic and biogenic atmospheric SOA.

The computational workflow set up for this proof of concept can be further extended to even more chemical classes and toxicological endpoints.

## 2.2 Toxicity endpoints

Standard *in silico* methods employed in pharmaceutics as well as in chemical risk assessment enable the prediction of a variety
of toxicological and eco-toxicological endpoints, such as skin and eye irritation, acute toxicity, genotoxicity, reproductive toxicity, aquatic toxicity etc. (Worth et al., 2011; Benfenati et al., 2016; ECHA 2016). The present study focuses only on genotoxicity, because a) it can be associated to any kind of exposure, b) it can be induced also at very small doses, c) it leads to adverse health effects on humans upon chronic exposures (and the increase in mortality risk for PM is much higher for chronic than for short-term exposures (Pope et al. 2011)), d) it is a well-studied endpoint with defined mechanisms of action
(Worth et al., 2011; Schultz et al., 2015). Most importantly, genotoxicity has been characterized in real PM samples and it is subject to monitoring in several countries (e.g., Cassoni et al., 2004). Past studies have attempted to apportion total PM mutagenicity among distinct primary organic aerosol sources (Hannigan et al., 2005). No such exercise has been carried out yet for SOA sources, in spite of the fact the mutagenicity of SOA have been demonstrated at least 30 years ago (Kamens et al., 1984). It is worth reminding that mutagenicity is only one of the possible toxicological endpoints for atmospheric
particulate organic compounds, and that the same classes of compounds can be more or less toxic depending on the specific endpoints (e.g., mutagenicity vs. cytotoxicity, etc.) (Filep et al., 2015).

In this study, QSAR predictions focus on Salmonella in vitro mutagenicity (Ames test). It should be noted that screening tests performed on strains of prokaryotes, such as the Ames test, are not meant to mimic exposure conditions for humans. They are used instead in the preliminary phases of hazard assessment, and especially for hazard identification.

## 2.3 Collection of experimental in vitro genotoxicity data

In this study, a search for in vitro mutagenicity (Ames test) data available for the 104 aerosol compounds was performed with the support of the OECD QSAR Toolbox (OECD 2009, 2013, 2015; detailed description of the tool is provided in SI - Table S2). The Toolbox is connected to a variety of toxicity databases and allows for gathering experimental data for the target compounds. The following databases were enquired for in vitro mutagenicity (Ames test) data: Bacterial mutagenicity ISSSTY
(by Istituto Superiore di Sanità), which includes TOXNET databases, Carcinogenic Potency Database (CPDB), ECHA CHEM, Toxicity Japan MHLW (EXCHEM), Genotoxicity OASIS, Carcinogenicity & Mutagenicity ISSCAN. A brief explanation of the screened databases is provided as Supporting Information (Table S3).

## 2.4 QSAR prediction of in vitro genotoxicity

The 104 aerosol compounds were then screened for their mutagenic potential by means of QSAR predictions. QSAR
predictions for Salmonella in vitro mutagenicity (Ames test) were generated by employing two models: i) ACD/Impurity Profiling model for Salmonella composite, implemented in the commercial software ACD/Labs Percepta (ACD/Percepta,

release 2015, www.acdlabs.com), and ii) Vega/CAESAR Mutagenicity model, implemented in the open platform VegaNIC (VegaNIC Application ver. 1.1.0, http://www.vega-qsar.eu/download.html). The two models, which fulfill the OECD principles for QSAR scientific validity, provide qualitative predictions for genotoxicity as Ames test (e.g., positive/negative) supported with specific parameters providing information on prediction reliability. Detailed information on the employed models are provided as Supporting Information (Table S4). The two QSAR models were used in order to complement each other. In case of conflicting reliable predictions, the most conservative health impact potential, i.e. positive prediction, was assigned.

## 2.5 Cluster analysis

A workflow for clustering the 104 aerosol compounds was designed with the aim to group compounds into clusters sharing similar physico-chemical, structural and mechanistic profile (Figure S1). The workflow was automatized using the KNIME Analytics Platform (Berthold et al., 2007) and is described below:

Step 1. Descriptors accounting for physico-chemical, molecular and reactivity properties: LogP (Log octanol-water partition coefficient), WS (water solubility), MW (molecular weight), TPSA (total polar surface area), nHBDon (number of donor atoms for hydrogen bonds), nHBAcc (number of acceptor atoms for hydrogen bonds), HOMO (highest occupied molecular orbital), LUMO (lowest unoccupied molecular orbital), etc…. These were calculated using PaDEL-Descriptors (Yap et al., 2011), ACD/Percepta, and Schrödinger software (Schrödinger Suite 2014 Update 3 Release, Maestro v9.9.013 - Sep 2014, evaluation copy). Bidimensional molecular descriptors (e.g., constitutional, topological, shape descriptors) were also calculated using PaDEL-Descriptors. Descriptors providing mechanistic information for genotoxic activity were derived using five genotoxicity profilers implemented in the OECD QSAR Toolbox (Table S2). In particular, the employed profilers identify in the target chemicals the presence of structural alerts for DNA binding or mutagenicity, with or without the associated MoA (mechanism of action) responsible for genotoxic activity. The outcome of the profilers was used to derive the mechanistic descriptors, which consisted in binary variables indicating the presence (1) or absence (0) of a structural alert (with or without MoA information). A total of 138 descriptors, including 18 physico-chemical descriptors, 70 structural descriptors and 50 mechanistic descriptors, was calculated. The complete list of calculated descriptors and their description is provided in Supporting Information (Table S5).

Step 2. To reduce redundant and non-useful information, constant descriptors (i.e., descriptors having the same value for all the compounds) and pair-correlated descriptors (i.e., descriptors with pair-wise correlation >0.8) among the physico-chemical and structural descriptors were removed. The initial set of 138 descriptors was reduced to 75 variables, including all 50 mechanistic descriptors, 8 PC descriptors and 17 structural descriptors (Table S5). Mechanistic descriptors were binary variables (0,1), while physico-chemical and structural descriptors were continuous variables. Thus, to analyse the three groups of descriptors in combination, continuous variables (i.e., physico-chemical and structural descriptors) were normalised within [0-1] range.

Step 3. A Principal Component Analysis (PCA) was performed on the final set of 75 descriptors to extract important information encoded by physico-chemical, structural and mechanistic properties and to reduce the number of input variables used for cluster analysis. The PCA was carried out in KNIME employing specific nodes for PCA (i.e., PCA compute and PCA apply), based on covariance matrix implementation method. The first 13 principal components (PCs) of the PCA were maintained, which preserved 85% of the information encoded in the original variables.

Step 4. Hierarchical clustering of the 104 compounds was performed using as input variables the first 13 principal components (PCs) of the PCA (see Step 3). An agglomerative clustering was performed employing the Hierarchical clustering KNIME node (metric: Euclidean distance; linkage criterion: average linkage). The number of clusters was set a priori (n = 10).

## 3 Results

### 3.1 Screening for genotoxicity – experimental data

Experimental data for Salmonella in vitro mutagenicity (Ames test) was available for 13 out of 104 aerosol compounds (i.e., ≈ 13%). These compounds include aromatic oxygenated compounds (found among the PAH-SOA constituents) and N-nitrosamines (formed by amine oxidation and NO addition). A summary of retrieved data is provided in Table 1. Detailed information on the retrieved experimental data are reported in Supporting Information (Table S6). Overall, it was noted that consistent negative experimental Ames test data (conducted on different Salmonella strains, with and without metabolic activation) are available for four compounds (two substituted benzene SOA compounds, and two from PAH-SOA). In addition, consistent positive experimental Ames test data (different Salmonella strains, with and without metabolic activation) are available for two compounds (one PAH-SOA compound, and one amine-SOA), while for the remaining seven compounds, both positive and negative Ames test results are reported, based on the Salmonella strain and testing conditions used.

### 3.2 Screening for genotoxicity – QSAR predicted data

*In silico* predictions for Salmonella in vitro mutagenicity were generated for the 104 studied aerosols employing two QSAR models, i.e. ACD/Percepta Impurity (Salmonella composite) model and Vega/CAESAR Mutagenicity model. Predictions obtained by the two QSARs were combined taking into account the applicability domain of each model and consistency between predictions (Table S7). Predictions were assessed for their reliability and different levels of confidence were assigned ("not", "borderline", "moderate" and "high" reliable predictions). Overall, the predictions generated by the two QSAR models were in agreement for the majority of compounds (not consistent predictions obtained for only 4 compounds out of the 53 reliably predicted by both QSAR models) and the combination of the two tools resulted successful in covering a wider chemical space (59% of compounds reliably predicted by ACD/Percepta alone, 74% of compounds reliably predicted by Vega/CAESAR alone, 82% of compounds reliably predicted combining the two QSAR models). As far the 4 compounds with opposite predictions are concerned, this is mainly due to issues related to the different applicability domain of the two QSAR models.

Based on the combined predictions, 53 compounds were predicted as negative, 32 as positive (including four compounds with conflicting reliable predictions), while for 19 compounds QSAR predictions were assessed as "indeterminate" (i.e., not reliable or equivocal predictions) (Table 2).

It is worth noting that 30% of the compounds were screened as genotoxic by at least one of the two QSAR models employed in this study. These genotoxic SOA components included all five N-nitrosamines from amine atmospheric oxidation, but none of the compounds chosen to represent isoprene-SOA, sesquiterpene-SOA or (among the anthropogenic) alkane photo-oxidation SOA. On the other hand, the relative large classes of compounds in the list correspondent to monoterpene-SOA, substituted benzene SOA and PAH-SOA included both several genotoxic and non-genotoxic species. In the following section, we provide clues for understanding inter- and intra-class variability.

## 3.3 Cluster analysis

The use of cluster analysis is two fold: 1) Identify genotoxicity trends within clusters, and 2) Identify gaps in experimental and *in silico* genotoxicity data. Hierarchical clustering initially resulted in 10 clusters, each characterised by a different composition in terms of number of compounds belonging to each cluster and the SOA class of each compound. Due to the high number of the members included in Cluster no. 9, an additional hierarchical clustering was performed on this cluster, which leads to four sub-clusters which are reported as such in proceeding analysis. Therefore, a total of 13 clusters were generated, as illustrated in Table S9. The 104 compounds are also visualised in the PCA score plots (Figure 1), where compounds are displayed in the space described by the first two principal components (PCs) and marked with different colours based on SOA classes and on clusters. The corresponding loading plot can be found in Figure S2.

The figure shows that SOA classes with compounds located in confined regions of the PCA score plots often correspond to individual clusters (e.g., amine-SOA with Cluster 2), while monoterpene-, substituted benzene- and PAH-SOA intercept more clusters and show diverse behaviours in the space described by the first two PCs. The distribution of compounds in the score plots of the first two PCs, which explain 35% of the total variance, is primarily based on the mechanistic descriptors. In particular, compounds with no or few structural alerts for genotoxicity (e.g., compounds belonging to cluster 9) are displayed in the bottom-right part of the plots (Figure S2), while the remaining compounds exhibit different alerts and mechanisms for genotoxicity. In particular, typical descriptors for the clusters characterized by high values of the second PC (clusters #7 and #8) are: radical reactivity via ROS formation, occurrence of hydroxyperoxides and of "H-acceptor-path3-H-acceptor" substructures (= two hydrogen acceptors at a distance of three atoms) (Figure S2). The role of other relevant descriptors (e.g., lipophilicity, electronic descriptors etc.) in the distribution of compounds among different clusters becomes more evident when analyzing the other principal components.

Within each cluster, SOA compounds were analysed in terms of structural and physico-chemical properties, mechanistic profile and mutagenic potential. The analysis and results for each cluster are provided in Supporting Information (Table S8a,b) but we present the overarching results in the following. In general, SOA classes (defined on the nature of precursors) show a variable degree of overlap with the clusters: there are some well populated clusters that include multiple SOA classes, in

particular sub cluster 9-3, including many aliphatic compounds from isoprene, dicarbonyls, monoterpene and substituted benzene oxidation, and which are mostly non mutagenic. On the other hand, there are distinct clusters of mutagenic or non-mutagenic compounds that are associated to specific SOA systems (Figure 2) (e.g., Cluster 1: mutagenic quinones from PAH oxidation, and sub Cluster 9-1: alkane-SOA, all non-mutagenic). By combining the information on compounds and properties from the clusters (encoding physico-chemical, structural and mechanistic descriptors of the compounds) with that of sources (specific SOA systems) (Table S9), we can list the following groups of mutagenic compounds (with characteristics of their mechanistic profile provided in parenthesis):

- Cluster 0: 5-membered-lactones in monoterpene SOA (alkylation via Michael-type addition of 5-membered-lactone as well as via ring opening SN2 reaction)

- Cluster 1: quinones from low MW PAHs oxidation (DNA intercalation, radical reaction mechanism via ROS formation, Michael addition to quinoid structure)

- Cluster 2: N-nitrosamines (nucleophilic attack after nitrenium or carbenium ion formation)

- Cluster 3: hydroxy- and nitro-PAHs (several mechanisms, including DNA intercalation, ROS formation, and nucleophilic attack after reduction and nitrenium ion formation). The same cluster includes one glyoxal-imidazole, for which no reliable QSAR prediction was obtained. For this compound experimental testing for mutagenicity is recommended.

- Cluster 7: non-aromatic products of substituted benzene oxidation (several mechanisms including ROS formation, alkylation by epoxides, and reactions involving a H-acceptor-path3-H-acceptor). This cluster also includes a formaldehyde -CI adduct formed by monoterpene ozonolysis..

- Cluster 8 (only 5/13 mutagenic): alkyl hydroperoxides and peroxyacids from substituted benzene, alkene and monoterpene oxidation (ROS generation, H-acceptor-path3-H-acceptor reactions), and one 5-membered-lactone in substituted benzene SOA (alkylation via Michael-type addition of 5-membered-lactone).

Other classes of mutagenic compounds were found within sub Cluster 9-2 (nitro- and oxo-PAHs) and sub Cluster 9-3 (only three compounds with heterogeneous descriptors). The abundance of mutagenic species in each cluster is graphically represented in Fig. 2 together with examples of molecular structures of representative mutagenic compounds in each of the major clusters and SOA types.

Note that not all compounds carrying genotoxicity alerts were predicted to be genotoxic from the QSARs. For instance, monoterpene-SOA dimers bearing ozonide bonds (Cluster 6) are likely precursors of ROS, but were predicted to be non-genotoxic with a moderate to high reliability. It is worth remembering that the mechanistic profile (e.g., the presence of reactive functional groups in a molecule) must be considered a collection of alerts, and that the mutagenicity depends on all the characteristics of the specific compound, which affect and modulate the actual expression of the alerts.

The six clusters of compounds listed above, accounting for 26 of the 32 compounds predicted mutagenic by the QSAR analysis, include three clusters representing groups of compounds with well-known toxicological properties: N-nitrosamines (Cluster 2), hydroxy- and nitro-PAHs (Cluster 3) and quinones (Cluster 1). These clusters encompass organic compounds with clear

source fingerprints (from PAHs and amine oxidation and reaction with NOx) and that are found in polluted environments in concentrations ranging from $10^{-3}$ to $10^0$ ng m$^{-3}$ (Chung et al., 2006; Huang et al., 2014; Farren et al. 2015) (Table S10). Nitro- and oxo-aromatic compounds are by far the best studied mutagenic components of SOA (e.g., Enya et al., 1997; Di Filippo et al. 2007; Huang et al., 2014). They can form upon oxidation of primary organic compounds which can be also mutagenic (PAHs) or that are just non toxic, such as metoxy-phenols in biomass burning emissions (Kroflič et al., 2015).

The other three clusters encompass oxygenated organic compounds that, in our knowledge, have been ignored so far in mutagenicity studies of atmospheric aerosol compounds. They could contribute to explain the yet unknown composition of the most polar fraction of mutagenic compounds in atmospheric aerosol (Barale et al., 1994; Gutiérrez-Castillo et al. 2006; Palacio et al., 2016). First, 5-membered lactones formed by the oxidation of monoterpenes are alkylating agent and possibly mutagenic (Cluster 0). These include terpenyllic acid, which occur in concentrations of 6 to 15 ng m$^{-3}$ in polluted rural areas (Kristensen et al.,2013; Brüggemann et al., 2017). A second class (Cluster 7) of mutagenic compounds encompass multi-functional oxygenated non-aromatic species (including epoxides) formed from the oxidation of substituted benzenes as well as the products of addition of formaldehyde to monoterpene Criegee Intermediates. Finally, a more heterogeneous class of SOA compounds originating from both biogenic (monoterpenes) and anthropogenic VOCs (substituted benzenes, alkenes) and characterized by the presence of certain functionalities like hydroperoxy-, peroxy-acidic groups or H-acceptor-path3-acceptor structures account for mutagenic compounds in Cluster 8. No experimental data are available for the ambient concentrations of organic compounds belonging to Clusters 7 or 8. However, SOA modelling studies indicate that they can account for a significant fraction of SOA mass formed by substituted benzene oxidation (Ruggeri et al., 2016), which translates into ambient concentrations of several ng m$^{-3}$ in the very polluted environments (such as in Yuan et al., 2013) (Table S10). The formation of endoperoxides and epoxides and of other reactive oxygenated organic compounds such as those in Clusters 7 and 8 was hypothesized by Fu et al. (2012) to explain the DNA damage caused by photodegraded aromatic compounds. Moreover, the finding of biogenic SOA components with potential mutagenic properties (Cluster 0, and some compounds in Clusters 7 and 8) is in agreement with the results of Alves et al. (2016), showing that mutagenicity of the aerosol is enhanced when pollution get mixed with emissions in rural areas. The overall picture, however, is that a greater proportion of mutagenic compounds was predicted to occur in the anthropogenic SOA classes then in the biogenic ones (Figure 2). Exceptions are alkane SOA which were predicted to be non-mutagenic, and some of the monoterpene SOA which instead showed mutagenic activity on the basis of our QSAR approach. These results show a very good, qualitative overlap with the findings of Verma et al. (2015) and Tuet et al. (2017) about the redox potentials of specific classes of SOA. Although ROS expression is only one of the possible mechanisms of action responsible for mutagenicity, it characterizes several of the compounds predicted to be mutagenic in our study, especially in Clusters 1, 3, 7 and 8. The formation of ROS in atmospheric SOA and the genotoxic effect of oxidative stress caused to PM exposure have been documented by a number of studies (e.g., Risom et al., 2005; Valavanidis et al. 2008; Rattanavaraha et al., 2011; Oh et al., 2011; Fu et al., 2012).

As a final remark, it should be noticed that the vast sets of molecular structures emerging from the laboratory experiments on the best-studied SOA systems (such as monoterpene and substituted benzene SOA) include compounds of very diverse

chemical and biological activity. As a corollary, it is possible that expanding the chemical characterization of the SOA components from other VOC precursors will lead to the identification of new potential mutagens. In summary, the identification of new classes of compounds with predicted mutagenic activity calls for more research research on the identification and quantification of such compounds in ambient aerosol samples, as well as on on the toxicological properties of the specific SOA systems, and to precautions in the laboratories involved in the synthesis of SOA tracer compounds (Nozière et al., 2015).

## 4 Conclusions

The present study represents a hitherto unexplored proof-of-concept investigation that *in silico* methods might be successfully employed to support safety assessment of atmospheric aerosol organic compounds through the screening and prioritization of the most hazardous ones based solely on the information of the chemical structure. In addition, the clustering of chemicals into small groups sharing common structural features, physico-chemical properties and mechanistic profiles, set the basis for the application of the read-across approach, where few representative compounds are tested and the outcome is extrapolated to similar compounds. The results, based on a list of ~100 individual organic compounds representative of the most important anthropogenic and biogenic SOA classes, indicate that diverse groupings of molecular species exerting specific genotoxic effects might be of important, and among these are classes of SOA compounds previously unexplored for their mutagenic potential. These includes 5-membered lactones and a heterogeneous class of oxygenated compounds with epoxide, peroxide functionalities and promoting ROS expression. Since, in our list of 104 compounds, redox-active species with mutagenic properties are more frequently found in SOA formed by the degradation of aromatic VOCs, the apportionment of genotoxic compounds among SOA classes shows an overall dominance of anthropogenic over biogenic species (Figure 2). Therefore these results, based on a first application of *in silico* methods on a limited number of molecular species, are in qualitative agreement with the existing general knowledge of aerosol health effects indicating a more robust association of combustion generated aerosols with adverse outcomes with respect to other aerosol sources (Delfino et al., 2009).

The usefulness of *in silico* methods stands in their ability in screening vast sets of aerosol organic substances. In principle this would include a much larger set than the 104 compounds considered here, whilst at the same time providing directions for implementing targeted in vitro (and in vivo) toxicological assessments on specific compounds. As already noted, there have been significant advances in atmospheric aerosol modelling and instrumentation. We cannot measure every compound in atmospheric aerosol. As mechanistic models have the potential to predict concentrations of many millions of compounds (McVat et al., 2016), even with uncertain process descriptions, the challenge is how to use this growing amount of information in a meaningful way. As we continually identify and hypothesize new processes and compounds deemed important for better understanding aerosol transformation and impacts, employing automated *in silico* methodologies offers the opportunity to replicate this proof of concept study with many simulations of a changing environment.

## Acknowledgments

This study was part funded by the National Environment Research Council in the UK (NERC) during project NE/J02175X/1.

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

**Table 1: Experimental data for Salmonella *in vitro* mutagenicity**

| Compound name | No. studies | Test Calls | Overall outcome[c] |
|---|---|---|---|
| 2-nitrophenol | 23 [a;b] | - (23) | - |
| 2-methylphenol | 20 [a;b] | - (20) | - |
| naphthol | 11 [a;b] | - (9), + (2) | + |
| naphthoquinone | 17 [a;b] | - (13), + (3), equivocal (1) | + |
| dihydroxy-phthalic acid | 27 [a;b] | - (27) | - |
| phenanthrene-9,10-dione | 6 [a;b] | - (6) | inconclusive |
| phenanthren-9-ol | 3 [a;b] | - (1), + (2) | + |
| 9-nitrophenanthrene | 16 [a;b] | - (2), + (14) | + |
| 4-nitro-6H-dibenzo[b,d]pyran-6-one | 6 [a;b] | - (5), + (1) | + |
| 1-nitro-pyrene | 25 [a;b] | - (1), + (19), equivocal (3) | + |
| N'-nitrosonornicotine | 18 [a;b] | - (13), + (4), equivocal (1) | + |
| 4-(methylnitrosoamino)-1-(3-pyridyl)-1-butanone | 7 [a;b] | - (3), + (3), equivocal (1) | + |
| 4-(Methylnitrosoamino)-1-(3-pyridyl)-1-butanol | 1 [a] | + (1) | + |

a Bacterial mutagenicity ISSSTY; b Genotoxicity OASIS; c Overall outcome according to Bacterial mutagenicity ISSSTY assessment: positive ("+") (at least one strain is positive (with or without Metabolic activation));  "equivocal" (no strain is positive, and at least one equivocal result is present in one of the following strains (with or without Metabolic activation): TA1535, TA100, TA98, TA1538, TA1535, TA97); negative ("-") (no positive or equivocal results are present in any strain, and negative outcomes exist for: a) at least one strain from among TA1535 or TA100 or TA97 (with and without Metabolic activation); and b) at least one strain from among TA1538 or TA98 or TA1537 (with and without Metabolic activation)); "inconclusive" (if none of the above criteria is fulfilled). Full information on genotoxity observation data (strains, metabolic activation, etc.) with references can be found in Table S6.

**Table 2: Results of the QSAR genotoxicity predictions from VEGA/Caesar and ACD/Percepta models.**

| | | VEGA/Caesar | | | | |
|---|---|---|---|---|---|---|
| | | Negative | Positive | out | n/a | tot |
| ACD/Percepta | Negative | 36 | 3 | 7 | - | 46 |
| | Positive | 1 | 13 | - | 1 | 15 |
| | Equivocal | - | 2 | 4 | - | 6 |
| | out | 10 | 12 | 14 | 1 | 37 |
| | tot | 47 | 30 | 25 | 2 | 104 |

Combining predictions obtained by the two models, the total number of compounds classified as genotoxic from either VEGA/Caesar or ACD/Percepta amounts to 32 species (red); the number of compounds determined as non-genotoxic by one model and that is not classified as genotoxic by the other is 53 (green); the number of compounds classified as indeterminate (i.e. "equivocal" or "not reliable" prediction)
5   (grey) by both methods is 19.

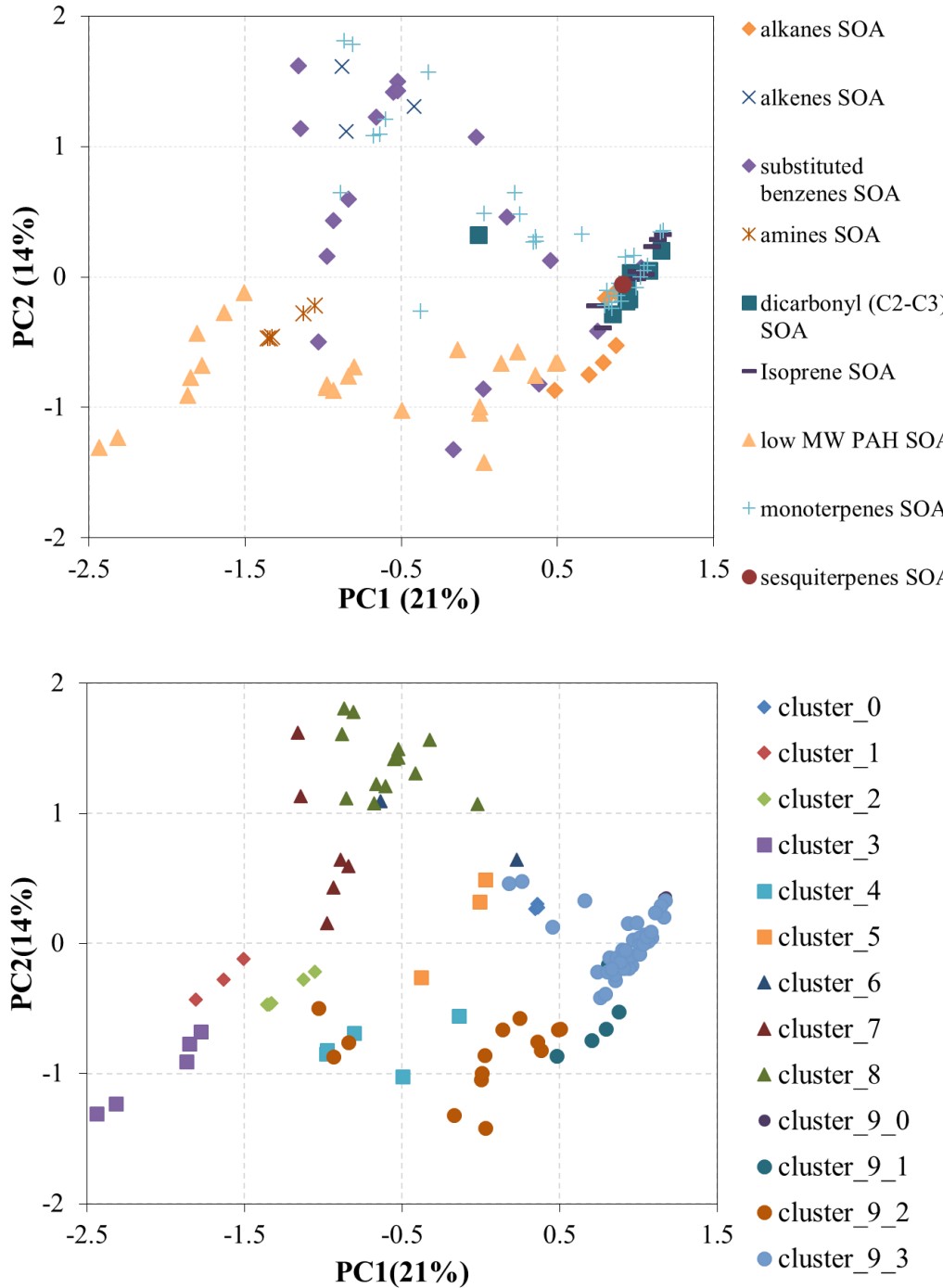

**Figure 1: PCA score plot of 104 aerosol compounds: compounds marked according to different SOA classes (top); compounds marked according to different clusters (bottom).**

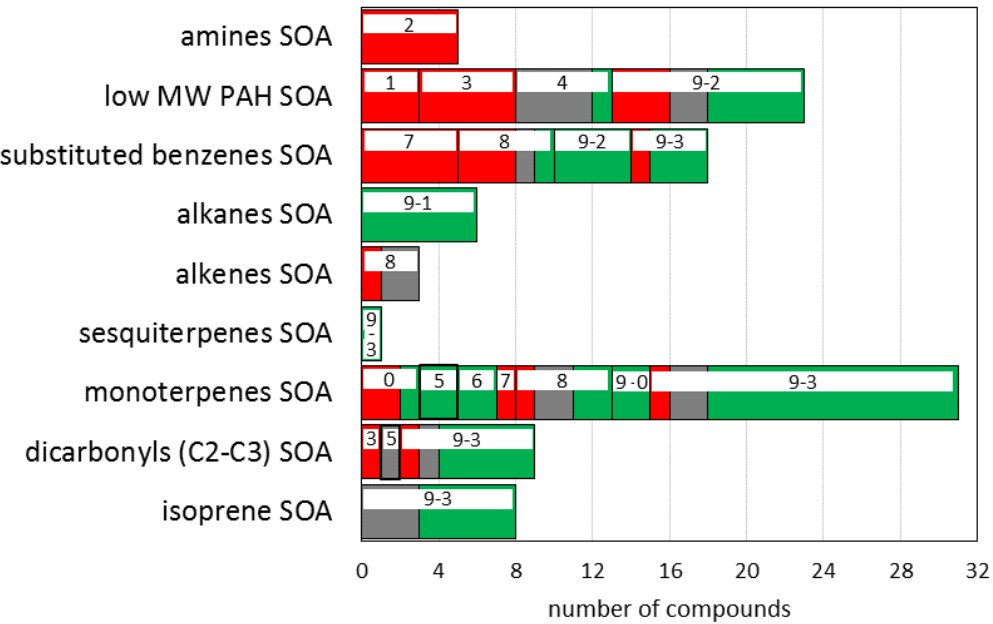

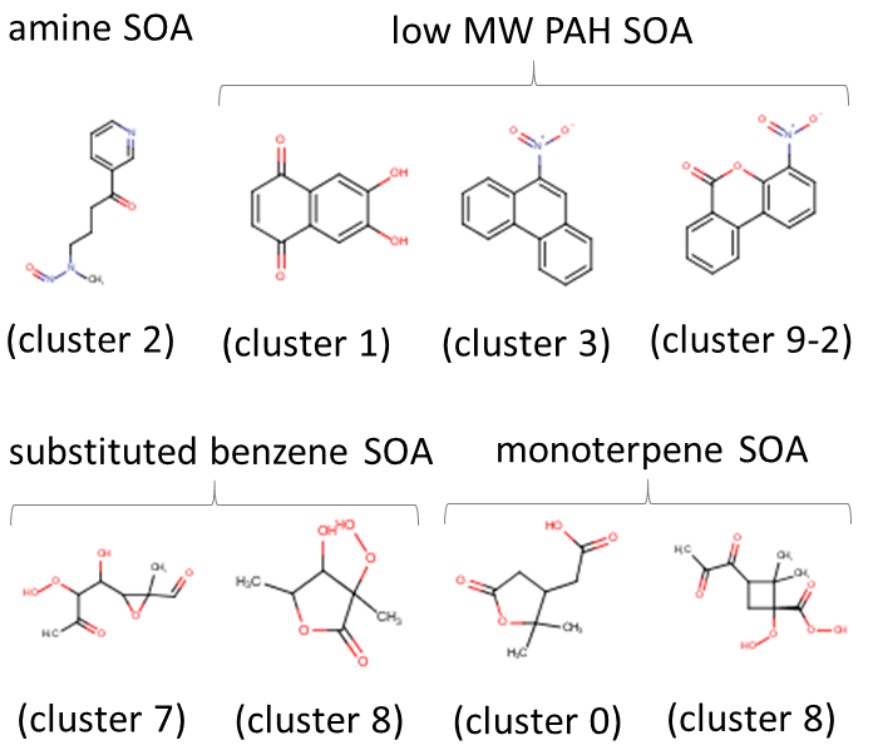

**Figure 2: (top) Distribution of mutagenic (red); non-mutagenic (green) and indeterminate (grey) compounds between clusters (labels in white boxes) and SOA classes. (bottom) Examples of molecular structures of compounds predicted to be mutagenic.**