# Peer review of "Evaluating the mutagenic potential of aerosol organic compounds using informatics based screening"

_Atmospheric Chemistry and Physics, 2017_

## Referee Comment (RC1) · Anonymous Referee #1 · 9 Aug 2017

This paper involves an analysis of existing data bases to infer possible toxic components of ambient SOA. The analysis reported is not comprehensive, serving mainly as a proof of concept. The idea is interesting, worthy of publication, but would be more appropriate for this journal if linked to existing knowledge on aerosol chemistry and health effects. My main suggestion with this paper is that a more substantial discussion should be added to put the work in context with known aerosol toxicity, source apportionment, and epidemiology studies. There is a substantial body of published studies that have identified various aerosol sources that are strongly linked to adverse health outcomes, such as incomplete combustion; eg, vehicle tail pipe emissions and biomass burning. These produce SOA and many of these compounds seem to be

found in the main factors of this study with high toxicity. In contrast, from other studies biogenic SOA tends to be less clearly associated with health effects (check out the published literature). So why not link Fig 2 in at least a qualitative way with published health studies through a more comprehensive discussion, possibly identifying factors or toxic compounds in this study linked to SOA from incomplete combustion, biogenic VOC SOA etc. Putting these results in a large context of published work would significantly increase the impact of these findings, at least for the readers of this journal.

Finally, it is exposure that determines health effects, that is toxicity times concentration of a given species. A discussion on this would also be very helpful. It would be useful if the authors could provide some idea of typical concentrations of these identified toxic species (or groups), maybe for a range of sites. For example, maybe only a small fraction of the biogenic SOA leads to a toxic substance, but maybe the concentration of these species are very high (or low) making it a potentially important (or not) species.

Minor comment. The format is a bit strange: Why two Introduction sections? Table 1 has no caption.

---

## Author Comment (AC1) · 8 Sep 2017

**REPLY TO REFEREE 1**

We thank the Referee for the useful comments. We agree that the current version of the manuscripts perhaps lacks a thorough recognition of previous studies on atmospheric aerosol health effects. We answer to his/her specific comments (reported here in *italics*) below:

*"This paper involves an analysis of existing data bases to infer possible toxic components of ambient SOA. The analysis reported is not comprehensive, serving mainly as a proof of concept. The idea is interesting, worthy of publication, but would be more appropriate for this journal if linked to existing knowledge on aerosol chemistry and health effects. My main suggestion with this paper is that a more substantial discussion should be added to put the work in context with known aerosol toxicity, source apportionment, and epidemiology studies. There is a substantial body of published studies that have identified various aerosol sources that are strongly linked to adverse health outcomes, such as incomplete combustion; eg, vehicle tail pipe emissions and biomass burning. These produce SOA and many of these compounds seem to be found in the main factors of this study with high toxicity. In contrast, from other studies biogenic SOA tends to be less clearly associated with health effects (check out the published literature). So why not link Fig 2 in at least a qualitative way with published health studies through a more comprehensive discussion, possibly identifying factors or toxic compounds in this study linked to SOA from incomplete combustion, biogenic VOC SOA etc. Putting these results in a large context of published work would significantly increase the impact of these findings, at least for the readers of this journal."*

**REPLY:** The Reviewer is right: there is massive amount of literature results for anthropogenic and natural aerosol health effects that is simply not referenced in our manuscript. We have addressed this, in part, by adding a range of references to cover studies highlighting distinct aerosol source contributions. On the other hand, we would like to note that much fewer studies have focused on SOA than on combustion aerosols, and mutagenicity is only one in the wide range of the possible endpoints explored in the field of aerosol toxicology. Moreover, a substantial fraction of recent studies focused on properties, such as ROS content or ROS expression, which cannot be considered strictly toxicological endpoints but more specifically refer to the mode of action of toxic agents. Even fewer studies performed source apportionment of the observed toxicity burden of the aerosol. To the best of our knowledge, the only study attempting source apportionment of ambient aerosol mutagenicity is the paper by Hannigan et al. (2005) which is based on a chemical mass balance (CMB) approach employing source profiles uniquely for primary organic aerosol, therefore providing no information on the potential contributions from SOA. The main body of the existing literature on role of SOA on aerosol mutagenic activity refers mostly to nitro-PAHs (and to a lesser extent, to oxo- and hydroxyl-PAHs) (e.g., Enya et al., 1997). References to compounds other than polyaromatics is quite sparse. To go back to the original point raised by the Reviewer, we thus provide below a list a papers dealing with mutagenic effects of SOA but not specifically or uniquely from polyaromatic compounds. We also included some of the recent studies of SOA toxicity targeting ROS expression, because this can be linked to the mechanistic profiles of some classes of oxygenated compounds predicted to be mutagenic by our QSAR approach. The overall picture certainly goes in the direction mentioned by the Reviewer: there is a more consistent evidence of the health effects for the products of anthropogenic combustion emissions than for other sources of organic aerosols (Delfino et al., 2009), although biogenic SOA systems have certainly been subjected to lesser extent. We will summarize these literature findings in a new dedicated paragraph to append to Section 3.3 and, in compact, form in the Conclusions.

Selected literature citations:

| Citation | Selected results: | Implications for the present study |
|---|---|---|
| Kamens et al., Environ. Sci. Technol., 18, 523, 1984. | - Mutagenicity in aged wood smoke increased of a factor of 2 to 10 with respect to fresh smoke. | - Experimental evidence of the mutagenic effects of SOA from combustion sources. |
| Alves et al., Environmental and Molecular Mutagenesis, 57, 41-50, 2016. | - Episodes of high mutagenicity in the Sao Paulo State (Brazil) seem to occur when the air mass from the rural area of sugar cane production is mixed with air in the region impacted by anthropogenic activities. | - Secondary organic reactions concomitant to both anthropogenic and biogenic SOA formation affect observed mutagenicity of ambient PM in rural areas. |
| Hannigan et al., Environ Health Perspect., 104, 428–436, 1996. | - Report stresses the importance of proximity to sources of direct emissions of bacterial mutagens but also implies that if 'important mutagen-forming atmospheric reactions occur, they likely occur in the winter and spring seasons as well as the photochemically more active summer and early fall periods.' | - Seasonal variation in mutagenicity, according to different atmospheric reactive conditions |
| Filep et al., Aerosol and Air Quality Research, 15: 2325–2331, 2015. | - Eco toxicity parameters (cyto and geno) are strongly emission source dependent; the higher the ratio of the biomass burning related carbonaceous aerosol the higher the cytotoxicity and the higher the ratio of the fossil fuel related carbonaceous aerosol the higher the genotoxicity. | - These results, obtained on biomass burning samples, indicate that specific mutagenic effects could change within one PM type – supports potential for composition dependency. |
| Kroflič et al., Scientific Reports **5**, Article number: 8859, 2015. | - assessment of the impact of low- and semi-volatile aromatic pollutants on the environment due to their atmospheric aqueous phase aging. It reveals that remote biotopes might be the most damaged by wet urban guaiacol-containing biomass burning aerosols. It is shown that only after the primary pollutant guaiacol has been consumed, its | - The study suggests pyrolysis of the polymer lignin during biomass burning leads to semi-volatile organic compounds (SVOC), such as guaiacol (GUA), which then produce nitro-aromatic derivatives. |

| | | |
|---|---|---|
| | probably most toxic nitroaromatic product is largely formed. | |
| Palacio et al., Mutation Research, 812, 1–11, 2016; Gutiérrez-Castillo et al., Environmental and Molecular Mutagenesis, 47, 199-211, 2006. | - Both organic and water-soluble associated compounds with particulate matter can produce genotoxic effects at concentrations commonly found in urban areas around the world | - Polar organic compounds accounting for SOA composition can contribute to the mutagenic activity of ambient PM. |
| Barale et al., Environ. Health Perspect., 102 (Suppl. 4), 67–73, 1994. | - Chemical fractionation of aerosol extracts showed that mutagenicity was contributed mostly by polar compounds not PAHs. | - Polar organic compounds accounting for SOA composition can contribute to the mutagenic activity of ambient PM. |
| Fu et al., Journal of Environmental Science and Health, Part C, 30, 1–41, 2012. | - Photo-oxidation products of polyaromatic comounds lead to the formation of epoxides, endoperoxides and quinones which eventually lead to ROS and DNA damage. | - Evidence for the mutagenic effect of aromatic oxygenated compounds including epoxides, endoperoxides and quinones as in several clusters of compounds predicted to be mutagenic in our paper. |
| Risom et al., Mutation Research, 592, 119–137, 2005; Oh et al., Mutation Research, 723, 142– 151, 2011. Valavanidis et al., Journal of Environmental Science and Health Part C, 26, 339–362, 2008. | - Oxidative stress caused by PM pollution is genotoxic;
 - Ambient air PM induces oxidative DNA damage in in vitro systems. | - Experimental evidence of PM-bound ROS-generating organic compounds (not limited to polyaromatic compounds) can be associated to mutagenicity. |
| Rattanavaraha et al., Atmospheric Environment, 45, 3848-3855, 2011 | - ROS expression in aged diesel emissions increased by a factor of 2-4 over fresh diesel particles.
 - The highest ROS potentials are found when including secondary organic aerosols from an $\alpha$-pinene, + toluene + an urban HC mixture. | - Several classes of compounds predicted to be mutagenic in our study are characterized by ROS expression among the alerts. Reaction chamber experiments have confirmed the production of redox-active secondary organic compounds in both biogenic and anthropogenic systems. |

| Tuet et al., Atmos. Chem. Phys., 17, 839–853, 2017. | - Redox-active compounds (measured by the DTT assay) are produced in negligible amounts in isoprene, caryophyllene and pentadecane SOA, in only moderate amounts in $\alpha$-pinene and monoaromatic SOA, and in much greater amounts in naphthalene SOA. | - Measured ROS expression in SOA qualitatively agree with the predicted mutagenicity of SOA tracers (Figure 2 of our paper), where isoprene, sesquiterpenes, *n*-alkane SOA compounds exhibit no mutagenic effects, while monoterpenes SOA include some mutagenic species and mostly non-mutagenic ones, while low MW PAHs SOA account for the largest share of mutagenic species. These results reinforce the link between ROS expression and mutagenicity. |
|---|---|---|
| Verma et al., Environ. Sci. Technol., 49, 4646–4656, 2015. | - Linear regression analysis between ROS generating capacity and OA fractions by AMS-PMF shows that in the SE USA, ROS expression of isoprene SOA is very small while biomass burning OA shows the greatest contribution. | - These results also agree qualitatively with the mutagenic activity of isoprene SOA compounds and of aromatic secondary species presented in our study. |

*"Finally, it is exposure that determines health effects, that is toxicity times concentration of a given species. A discussion on this would also be very helpful. It would be useful if the authors could provide some idea of typical concentrations of these identified toxic species (or groups), maybe for a range of sites. For example, maybe only a small fraction of the biogenic SOA leads to a toxic substance, but maybe the concentration of these species are very high (or low) making it a potentially important (or not) species."*

REPLY: Most of the SOA studies quoted in Table S1 focus on the identification of specific organic compounds on the basis of mass spectrometric analyses. However, the paucity of authentic standards makes quantitation challenging. As a result, the literature provides rather sparse information on the actual abundance of SOA markers in atmospheric samples. Most consistent data refer to well-studied systems such as $\alpha$-pinene SOA and isoprene SOA. For instance, Kristensen et al. (2014) found that the most common $\alpha$-pinene SOA tracers, including pinonic, hydroxyl-pinonic and pinic acids (mtr_05, mtr_06 and mtr_07 in Table S1), terpenyllic, terebic and diaterpenyllic acids (mtr_11, mtr_12 and mtr_13), 3-methyl-1,2,3-butanetricarboxylic (MBTCA) (mtr_17), together with their dimers (mtr_15, mtr_16), overall account for 10 to 15% of SOA mass. Similarly, Lin et al (2013) showed that isoprene SOA tracers can represent 12 – 14% of ambient particulate organic matter in an environment characterized by strong isoprene emissions. The contributions of the diverse isoprene SOA species varied a lot, with much greater shares from methyl-tetrols (iso_01 in Table S1), C5-alkene triols (iso_04) and their sulfate esters (iso_02) than from 2-methyl-glyceric acid and its derivatives (iso_06, iso_07). No concentration data are available for the majority of the compounds listed in Table S1, making any attempts of exposure assessment impossible for them. We would like to clarify, however, that the toxicity predictions conducted in our study are useful mainly for *hazard identification*, which is only the first step of risk assessment. Clearly, additional information on concentrations and exposure, as well as

compound-specific dose-response functions are required to characterize the health risk associated with the SOA compounds predicted to be mutagenic in this study. On the other hand, hazard identification is necessary to guide the subsequent steps of risk assessment, including the development of adequate analytical techniques to determine the concentrations of specific chemical compounds according to a priority list.

*"Minor comment. The format is a bit strange: Why two Introduction sections? Table 1 has no caption."*

**REPLY:** That is an error: Section 2 is in fact "Methods" not again "Introduction". It is also true that Table 1 has only footnotes not a proper caption. The table reports a summary of the observed mutagenic properties of the 13 organic compounds for which experimental data could be found in the literature.

**REFERENCES:**

Delfino et al., Air Pollution Exposures and Circulating Biomarkers of Effect in a Susceptible Population: Clues to Potential Causal Component Mixtures and Mechanisms, Environ. Health. Perspect., 117, 152-156, 2009.

Enya et al., 3-Nitrobenzanthrone, a Powerful Bacterial Mutagen and Suspected Human Carcinogen Found in Diesel Exhaust and Airborne Particulates, Environ. Sci. Technol., 31 (10), 2772–2776, 1997.

Hannigan et al., Source Contributions to the Mutagenicity of Urban Particulate Air Pollution, J. Air & Waste Manage. Assoc. 55, 399-410, 2005.

Kristensen et al., Dimers in $\alpha$-pinene secondary organic aerosol: effect of hydroxyl radical, ozone, relative humidity and aerosol acidity, Atmos. Chem. Phys., 14, 4201–4218, 2014.

Lin et al., Investigating the influences of $SO_2$ and $NH_3$ levels on isoprene-derived secondary organic aerosol formation using conditional sampling approaches, Atmos. Chem. Phys. 13, 8457–8470, 2013.

---

## Referee Comment (RC2) · Anonymous Referee #2 · 15 Sep 2017

The authors present a proof of concept study for the evaluation of the mutagenic potential of species in secondary organic aerosol. By screening 104 different compounds for mutagenicity potential using in silico toxicological predictions, the authors were able to suggest groups of compounds as well as specific molecules which should undergo toxicological screening in the future. This methodology could lead to a more focused approach when investigating the health impacts of different types of SOA by relating SOA composition to potential health impacts. This is an interesting manuscript which is generally well written and should be published after some revisions.

Comments: - Have the compounds which were tested previously been measured in
atmospheric SOA, and if they have at what concentrations and which types of SOA (e.g. anthropogenic, biogenic, etc)? If they haven't been specifically measured can the authors give a rough estimation of the concentrations that could be expected in different types of SOA? Is there a trend of more of the toxic compounds being likely to be present in a specific type of SOA?

- Are the concentrations which would be expected in the respiratory tract upon inhalation of atmospheric SOA similar to the concentrations used during the Ames test and assumed in the models? When both positive and negative Ames tests results are reported due to testing conditions (Page 7, Line 2) are there any trends that have been observed for the range of results (e.g. a dependence on concentration, or a different experimental condition) and could some of the experimental results be discarded due to those conditions being unlikely to occur in the respiratory tract/ human body after inhalation?

- Are there any limitations to the two models used (the ACD/Impurity Profiling model and the Vega/CAESER model), for example, can they reliably predict the toxicity of any compound? What are the differences between them which leads to one giving a positive result whilst the other gives a negative result in some cases (as shown in Table 2)?

Minor comments:

- Introduction line 24: Did the author mean nearly three million deaths per year (rather than billion)? This should also include a reference.

- There are a lot of abbreviations throughout the text which should be written out in full for clarity (e.g. PaDEL, nHBDon, nHBAcc, KNIME).

- Section 2 should be renamed from 'Introduction' to 'Methods'.

---

## Author Comment (AC2) · 3 Oct 2017

REPLY TO REFEREE'S 2 COMMENTS

We thank the Referee for his/her useful comments which give us the opportunity to clarify some aspects of the methodology.

COMMENT: Have the compounds which were tested previously been measured in atmospheric SOA, and if they have at what concentrations and which types of SOA (e.g. anthropogenic, biogenic, etc)? If they haven't been specifically measured can the authors give a rough estimation of the concentrations that could be expected in

different types of SOA? Is there a trend of more of the toxic compounds being likely to be present in a specific type of SOA?

REPLY: We have screened the existing literature and compiled a new table (Table S10, in attach to the present document, and to be included in the Supplementary Information of the revised paper) with concentration data observed for a subset of SOA compounds from the most relevant clusters of mutagenic species in our list. When direct observations are not available, we provided a rough estimate of ambient concentrations on the basis of observed/predicted yields in laboratory setups and on source apportionment results for ambient organic aerosols. We found that the new SOA compounds predicted to be mutagenic can occur in ambient air in concentrations of 10E-2 to 10E1 ng m-3. In respect to the Referee's question about possible trends in the abundance of toxic compounds between different classes of SOA, we think that Figure 2 of the present version of the manuscript already addresses this. The figure shows that a greater fraction of species predicted to be mutagenic is found in anthropogenic SOA systems than in the natural ones. As observed by Referee 1, this finding - though limited to the pool of compounds considered in our study - is qualitatively in line with the current evidence of clearer toxicological and epidemiological effects of anthropogenic combustion-related aerosol with respect to biogenic particles. We will add a note about this in the Discussion section of the revised version of the manuscript.

COMMENT: Are the concentrations which would be expected in the respiratory tract upon inhalation of atmospheric SOA similar to the concentrations used during the Ames test and assumed in the models? When both positive and negative Ames tests results are reported due to testing conditions (Page 7, Line 2) are there any trends that have been observed for the range of results (e.g. a dependence on concentration, or a different experimental condition) and could some of the experimental results be discarded due to those conditions being unlikely to occur in the respiratory tract/ human body after inhalation?

REPLY: We are happy to re-iterate and further clarify the role of presented data in our

study using three key points related to the questions posed.

1) First of all, it is worth re-iterating that mutagenicity is often related to long-term, chronic exposures in environmental toxicology. Even trace air concentrations of mutagens cumulate to mg-level doses during a lifetime exposure. That is why the WHO air quality guidelines recommend sub-ng/m3 threshold values for the concentrations of benzo[a]pyrene (WHO, 2000). The concentrations of the new SOA compounds predicted to be mutagenic in the Ames test on the basis of our QSAR approach are not always known. The ranges of concentrations of the compounds predicted to be mutagenic in our study (new Table S10, in attach to the present document) compare well with those observed for known atmospheric organic mutagens such as PAHs and nitro-PAHs (Alves et al., 2017). In conclusion, the newly identified mutagenic compounds can occur in ambient air in appreciable concentrations and be inhaled in mg amounts during a lifetime exposure. We therefore recommend them for in vitro toxicological screening for confirmation of their mutagenic effect and for determination of dose-response functions.

2) As far as the comparison with the doses used in the Ames test is concerned, we would like to emphasize that the Ames test (or bacterial reverse mutation test) is only a screening in vitro test performed to support a preliminary hazard assessment of chemicals, which are screened for their mutagenic potential. In more detail, the Ames test uses amino-acid requiring strains of Salmonella typhimurium and Escherichia coli to detect point mutations (OECD, 1997). Such mutations lead to revertant bacteria in which the functional capability to synthesize the essential amino acid is restored. Revertant bacteria are then detected by their ability to grow in absence of the amino acid necessary for the growth of the parent test strain. A wide range of concentrations are tested, with an upper limit which mainly depends on the solubility and cytotoxicity of the test compound in the final treatment mixture. The tested concentrations are expressed in either $\mu$g/plate or $\mu$L/plate, where "plates" are samples of bacterial strains, which makes a direct comparison with exposure concentrations for humans not properly feasible. It is important to emphasize that screening tests, such as the Ames test, are supposed to be used in the preliminary phases of hazard assessment, and especially for hazard identification, and not to mimic exposure conditions. We will add this to the text of the new version of the manuscript. Finally, as for the appropriateness of conditions used in models to replicate conditions in the respiratory tract, this is not within scope of the paper. Indeed, as we iterate, the driver for this work is to demonstrate the efficacy of a methodology to highlight potential hazardous compounds that might be measured in future epidemiological studies.

3) The variation of "testing conditions" mentioned in the manuscript includes factors such as the testing methodology (e.g., plate incorporation method or preincubation method), the tested strains of Salmonella typhimurium and Escherichia coli (at least five different strains are required, as specified in the OECD guideline, in order to detect different mutations), presence or absence of metabolic activation, exposure concentration. No specific trends have been observed among positive and negative results. The Ames test is an in vitro test using prokaryotic cells and, as such, it does not consider important processes in organisms of higher clades (e.g., rodent in vivo toxicological studies). Therefore, there are no experimental conditions that can be considered representative for the mechanisms of uptake of the test substance by humans (e.g., absorption through the respiratory tract), and, as a corollary, no specific testing conditions used in the Ames test could be discarded.

COMMENT: Are there any limitations to the two models used (the ACD/Impurity Profiling model and the Vega/CAESER model), for example, can they reliably predict the toxicity of any compound? What are the differences between them which leads to one giving a positive result whilst the other gives a negative result in some cases (as shown in Table 2)?

REPLY: This is a good point and we are happy to clarify here. All QSAR models are characterized by a defined applicability domain which means that no QSAR model can be guaranteed to provide a reliable prediction for any compound, regardless of

the endpoint. In this specific case, both the ACD/Impurity Profiling QSAR model and Vega/CAESER model are supported with specific parameters providing information on prediction reliability, including applicability domain considerations. As described in the Supplementary Information (Table S4), ACD/Percepta predictions are supplemented with a Reliability Index (RI), which ranges from 0 to 1, and gives an evaluation of whether a submitted compound falls within the Model Applicability Domain. In particular: RI < 0.3 (Not Reliable), RI in range 0.3-0.5 (Borderline Reliability), RI in range 0.5-0.75 (Moderate Reliability), RI >= 0.75 (High Reliability). Estimation of the RI takes into account two main aspects: similarity of the tested compound to the training set and the consistency of experimental values for similar compounds. Similarly, Vega/CAESAR predictions are provided with an Applicability Domain Index (ADI), which ranges from 0 to 1, and gives an evaluation of whether a submitted compound falls within the Model Applicability Domain. In particular: ADI > 0.9 means that the predicted substance is into the AD of the model; ADI < 0.7 means that the predicted substance is out of the AD of the model; ADI in range 0.7-0.9 means that the predicted substance could be out of the AD of the model and further considerations are needed. The ADI is calculated by grouping several other indices, each one taking into account a particular issue, such as: training/test set similar molecules with known experimental value, concordance for similar molecules, accuracy of prediction for similar molecules, atom Centered Fragments similarity check and model descriptors range check.

Predictions obtained by the two QSARs were combined by taking into account the applicability domain of each model (prediction reliability) and consistency between predictions. In case of conflicting reliable predictions, the most conservative outcome, i.e. positive prediction, was assigned. We will add the following text to the revised version of the manuscript:

"Overall, the predictions generated by the two QSAR models were in agreement for the majority of compounds (not consistent predictions obtained for only 4 compounds out of the 53 reliably predicted by both QSAR models) and the combination of the

two tools resulted successful in covering a wider chemical space (59% of compounds reliably predicted by ACD/Percepta alone, 74% of compounds reliably predicted by Vega/CAESAR alone, 82% of compounds reliably predicted combining the two QSAR models). As far the 4 compounds with opposite predictions are concerned, this is mainly due to issues related to the different applicability domain of the two QSAR models."

Minor comments (1): There are a lot of abbreviations throughout the text which should be written out in full for clarity (e.g. PaDEL, nHBDon, nHBAcc, KNIME).

REPLY: These includes names of softwares (PaDEL-Descriptor, and KNIME) for which specific references are already reported in the manuscript. We will instead report explanations for the following acronyms: nHBDon (= number of donor atoms for hydrogen bonds) and nHBAcc (= number of acceptor atoms for hydrogen bonds).

Minor comments (2): There - Section 2 should be renamed from 'Introduction' to 'Methods'.

REPLY: Yes, we thank the Referee for noticing this mistake.

REFERENCES:

1) Alves et al., "Polycyclic aromatic hydrocarbons and their derivatives (nitro-PAHs, oxygenated PAHs, and azaarenes) in PM2.5 from Southern European cities", Science of the Total Environment 595 (2017) 494–504.

2) OECD GUIDELINE FOR TESTING OF CHEMICALS n. 471. Bacterial Reverse Mutation Test. Available at: http://www.oecd-ilibrary.org/environment/test-no-471-bacterial-reverse-mutation-test_9789264071247-en.

3) WHO (2000) Air Quality guidelines for Europe, 2nd edition: WHO Regional Office for Europe (WHO Regional Publications, European Series, No. 91).

Please also note the supplement to this comment:

https://www.atmos-chem-phys-discuss.net/acp-2017-574/acp-2017-574-AC2-supplement.pdf

[Figure]

**Supplement:**

**Table S10**

| Cluster | ID | molecule | Common name | Precursor | Measured ambient concentrations of SOA compounds (ng/m³). | Estimated SOA *class* concentrations (ng/m³) in ambient air. | % compound in SOA mass (w/w) in reaction chamber conditions. | Calculated ambient concentrations of SOA compounds (ng/m³). ⊥ |
|---|---|---|---|---|---|---|---|---|
| 0 | mtr_11 |  | terpenylic acid | α-pinene and other monoterpenes | 6 – 15 [1, 2] | | | |
| 1 | lmp_17 |  | phenanthrene-1,4-dione | phenanthrene | 0.3 – 1 [3] | | | |
| | lmp_02 |  | naphthoquinone | naphthalene | 0.06 – 0.15 [3] | | | |
| 2 | ara_05 |  | N'-nitrosoanatabine | nicotine | 0.16 – 0.18 [4] | | | |

| | | | | | | | | |
|---|---|---|---|---|---|---|---|---|
| | ara_02 |  | 4-(methylnitrosoamino)-1-(3-pyridyl)-1-butanone | nicotine | 0.29 – 0.57 [4] | | | |
| 3 | lmp_21 |  | 9-nitrophenanthrene | phenanthrene | 0.003 – 0.019 [5,6] | | | |
| | lmp_23 |  | 1-nitro-pyrene | pyrene | 0.005 – 0.016 [5,6] | | | |
| | lmp_10 |  | 4-nitro-naphthol | naphthalene | | 171 – 276 (total naphthalene SOA) [7]§ | 0.4% [8] | ~ 0.7 -1 |
| | lmp_11 |  | | naphthalene | | 171 – 276 (total naphthalene SOA) [7]§ | 0.39% [8] | ~ 0.7 -1 |
| 7 | alb_09 |  | | 1,3,5-trimethyl-benzene (TMB) | | 37 (total TMB SOA) [9]* | ~ 5% [10]φ | ~ 2 |

| No. | ID | Structure | | | | SOA class conc. (ng/m³) | SOA mass fraction | Calculated ambient conc.⊥ |
|---|---|---|---|---|---|---|---|---|
| 8 | alb_10 |  | | 1,3,5-trimethyl-benzene (TMB) | | 37 (total TMB SOA) [9] * | ~ 2% [10] φ | ~ 0.7 |
| | alb_8 |  | | 1,3,5-trimethyl-benzene (TMB) | | 37 (total TMB SOA) [9] * | ~ 55% [10] φ | ~ 20 |
| 9_2 | lmp_12 |  | | naphthalene | | 171 – 276 (total naphthalene SOA) [7]§ | 0.07% [8] | ~ 0.1 – 0.2 |
| 9_3 | dic_01 |  | glyoxal | dicarbonyls | 0.8 – 2.7 [11, 12] | | | |

⊥ Calculated ambient concentrations of SOA compounds are provided when direct field observations are not available, by multiplying compound SOA mass fractions determined in reaction chamber experiments (second column from the right) with estimated SOA *class* concentrations (ng/m³) determined in ambient air by organic source apportionment methods (third column from the right).

§Total naphthalene SOA concentrations were derived using a molecular tracer method.

*Total TMB SOA concentrations were estimated from the calculated SOA formation yields expressed as µg/m³ of SOA mass per ppm of carbon monoxide (CO) (Table 4 in Yuan et al. 2013, low-NOx conditions) multiplied by a campaign-average CO concentration of 0.55 ppm.

φ Total TMB SOA Compound SOA fractions (Figure 9 in the paper of Ruggeri et al. 2016) result from the application of a gas-phase VOC oxidation model (Master Chemical Mechanism v3.2) coupled to a partitioning model (SIMPOL 1).

References:

1. Kristensen et al., Formation and occurrence of dimer esters of pinene oxidation products in atmospheric aerosols, Atmos. Chem. Phys., 13, 3763–3776, 2013;
2. Brüggemann et al., Real-time detection of highly oxidized organosulfates and BSOA marker compounds during the F-BEACh 2014 field study, Atmos. Chem. Phys., 17, 1453–1469, 2017;
3. Chung et al., Aerosol-borne quinones and reactive oxygen species generation by particulate matter extracts, Environ. Sci. Technol., 40, 4880-4886. 2006;
4. Farren et al., Estimated exposure risks from carcinogenic nitrosamines in urban airborne particulate matter, Environ. Sci. Technol., 49, 9648–9656, 2015;
5. Di Filippo et al., Air pollutants and the characterization of the organic content of aerosol particles in a mixed industrial/semi-rural area in central Italy, J. Environ. Monit., 9, 275–282, 2007;
6. Huang et al., Phase distribution, sources and risk assessment of PAHs, NPAHs and OPAHs in a rural site of Pearl River Delta region, China, Atmos. Poll. Res., 5, 210 – 218, 2014;
7. Lewandowski et al., Secondary organic aerosol characterization at field sites across the United States during the spring-summer period, Int. J. Environ. Anal. Chem., 93, 1084 – 1103, 2013;
8. Kautzman et al., Chemical composition of gas- and aerosol-phase products from the photooxidation of naphthalene, J. Phys. Chem. A, 114, 913–934, 2010;
9. Yuan et al., VOC emissions, evolutions and contributions to SOA formation at a receptor site in eastern China, Atmos. Chem. Phys., 13, 8815–8832, 2013;
10. Ruggeri et al., Model–measurement comparison of functional group abundance in $\alpha$-pinene and 1,3,5-trimethylbenzene secondary organic aerosol formation, Atmos. Chem. Phys., 16, 8729–8747, 2016;
11. Ho et al., Dicarboxylic acids, ketocarboxylic acids, and dicarbonyls in the urban atmosphere of China, J. Geophys. Res. A, 112, D22S27, doi:10.1029/2006JD008011, 2007;
12. Kampf et al., Development and validation of a selective HPLC-ESI-MS/MS method for the quantification of glyoxal and methylglyoxal in atmospheric aerosols (PM2.5), Anal. Bioanal. Chem., 401, 3115–3124, 2011.